# pH-dependent activation of cytokinesis modulates *Escherichia coli* cell size

**Elizabeth A. Mueller**, **Corey S. Westfall**, **Petra Anne Levin***

Department of Biology, Washington University in St. Louis, St. Louis, Missouri, United States of America

* plevin@wustl.edu

## Abstract

Cell size is a complex trait, derived from both genetic and environmental factors. Environmental determinants of bacterial cell size identified to date primarily target assembly of cytosolic components of the cell division machinery. Whether certain environmental cues also impact cell size through changes in the assembly or activity of extracytoplasmic division proteins remains an open question. Here, we identify extracellular pH as a modulator of cell division and a significant determinant of cell size across evolutionarily distant bacterial species. In the Gram-negative model organism *Escherichia coli*, our data indicate environmental pH impacts the length at which cells divide by altering the ability of the terminal cell division protein FtsN to localize to the cytokinetic ring where it activates division. Acidic environments lead to enrichment of FtsN at the septum and activation of division at a reduced cell length. Alkaline pH inhibits FtsN localization and suppresses division activation. Altogether, our work reveals a previously unappreciated role for pH in bacterial cell size control.

## Author summary

Bacteria are constantly under assault from endogenous and environmental stressors. To ensure viability and reproductive fitness, many bacteria alter their growth and replication in response to stressful conditions. Previous work from many groups has identified regulatory mechanisms linking cell division with nutrient availability and metabolic state. However, comparatively little is known about how the cell division machinery responds to physical and chemical cues in the environment. Here, we identify a fundamental property of the extracellular environment—environmental pH—as a significant contributor to bacterial cell size. Our genetic and cytological data indicate pH-dependent changes in *E. coli* cell size are in part due to differential localization of the cell division activator FtsN across pH environments. Increased abundance of FtsN at midcell in acidic environments promotes cell division at a reduced cell volume, while decreased abundance of FtsN at midcell in alkaline environments effectively delays cell division until a larger size is reached. Altogether, our work identifies pH as an environmental determinant of *E. coli* cell division and illuminates FtsN recruitment as a mediator of cell size.

**Data Availability Statement:** All relevant data are within the manuscript and its Supporting Information files.

**Funding:** This work was supported by National Institutes of Health grant GM127331 to PAL

(https://www.nigms.nih.gov), an Arnold O. Beckman postdoctoral fellowship to CSW (http://www.beckman-foundation.org/programs/beckman-postdoctoral-fellows), National Science Foundation graduate research fellowship DGE-1745038 to EAM (https://www.nsfgrfp.org), and a Center for Science and Engineering of Living Systems graduate scholar fellowship to EAM (https://livingsystems.wustl.edu). The funders had no role in study design, data collection and analysis, decision to publish, or preparation of the manuscript.

**Competing interests:** The authors have declared that no competing interests exist.

## Introduction

Size is a fundamental property of cells and is tightly linked to physiological state. With few exceptions, two processes dictate cell size: cell growth and cell cycle progression. During steady state or "balanced" growth, bacteria add on average the same volume between cell birth and division regardless of their size at birth. This phenomenon, referred to as the 'adder' model for bacterial cell size homeostasis, results in convergence to an average cell size [1,2]. Simulations and experimental data suggest that adder is an emergent property of two processes: 1) growth rate-dependent synthesis of rate-limiting components of the cell division machinery and 2) accumulation of these proteins to threshold numbers necessary to support cytokinesis. Consistent with this model, perturbing accumulation of one such protein, the tubulin homolog FtsZ, disrupts the volume added between divisions. Altering the onset of DNA replication fails to disrupt homeostatic cell size. Therefore, cell division—and not cell cycle progression generally—ultimately controls the volume of new material cells add during steady state growth [3].

Although there is little variation in size during steady state growth under a single, constant condition, changes in the environment can drastically affect the average cell size of single celled organisms. Nutrient availability, in particular, has a dramatic and positive impact on the size of evolutionarily distant bacteria—including *Escherichia coli*, *Salmonella enterica*, and *Bacillus subtilis*—as well as on budding and fission yeast [4–6]. Bacteria undergo a three-fold increase in cell volume when cultured in nutrient-rich conditions as compared to nutrient-poor conditions. Cell length and cell width both scale with nutrient availability in *E. coli* [7], while width remains nearly constant for *B. subtilis* [8]. The molecular basis of the positive relationship between nutrients and cell size is multifactorial, involving changes in biosynthetic processes that underlie cell growth [9,10] and the pathways mediating cell cycle progression [4,5,11,12]. Notably, nutrient-dependent changes in cell cycle progression identified to date all impinge on FtsZ assembly. In *B. subtilis* and *E. coli*, accumulation of the metabolite uridine disphosphate (UDP)-glucose during growth in carbon-rich media activates two unrelated glucosyltransferases, UgtP and OpgH, which antagonize FtsZ ring assembly. Although mechanistically distinct, both antagonists functionally increase the threshold quantity of FtsZ that must accumulate prior to cytokinesis [4,5].

While regulatory mechanisms coordinating division, nutrient availability, and size are well documented, comparatively little is known about how the cell division apparatus responds to other environmental cues. The division machinery in *E. coli* consists of over 20 proteins, collectively referred to as the divisome. These proteins assemble in a hierarchical fashion, beginning with midcell polymerization of FtsZ in the cytosol and ending with recruitment of septal cell wall synthesis enzymes and their regulators in the periplasm [13,14]. Similar to true extracellular processes, the periplasm is sensitive to changes in the abundance of ions and other small molecules due to the semi-permeable outer membrane and assumes the environmental pH while the cytosol remains relatively buffered at steady state [15–17]. Thus, the division proteins with domains in the periplasm are directly exposed to dynamic and potentially extreme environmental conditions—including changes in pH, osmolarity, and ionic strength—that may impact their ability to activate and complete cross wall synthesis. Differential activation of periplasmic components of the cell division machinery is sufficient to alter cell size at steady state: in *E. coli* and *Caulobacter crescentus*, gain-of-function mutations that affect the initiation or rate of septal cell wall synthesis consistently reduce size independent of changes to growth rate [18–22]. However, whether extracytoplasmic division proteins represent native integration points for environmental modulation of cell size remains unclear.

Here, we identify environmental pH as a conserved, growth rate-independent determinant of cell size in evolutionarily distant bacterial species. Distinct from nutrient-dependent

changes in size, which stem from changes in FtsZ assembly, pH predominately affects periplasmic components of the division machinery. Specifically, pH-dependent changes in *E. coli* cell length appear to stem from differential localization of the terminal division protein and cell wall synthesis activator, FtsN. Collectively, our data support a model in which pH-dependent changes in accumulation of FtsN at the cytokinetic ring impact the volume at which cells initiate division, thereby altering average cell size.

## Results

### pH influences cell size in diverse bacteria

To investigate the impact of pH on cell size, we cultured *E. coli* strain MG1655 at steady state in nutrient-rich media (LB + 0.2% glucose) under a physiologically relevant range of pH conditions (pH 4.5–8.5) [23,24]. We sampled cultures and fixed the cells during early exponential phase ($OD_{600} \sim 0.1$–0.2) for cell size analysis. At this time point, the pH of the culture had not significantly deviated from the starting pH (S1 Fig). Strikingly, cells cultured at pH 4.5 were ~75% of the area of their counterparts grown at pH 7.0. In contrast, cells cultured at pH 8.5 were ~120% of the area of cells grown at pH 7.0 (Fig 1A and 1B; S2 Fig and S2 Table). For comparison, loss of the metabolic cell size regulators *opgH* and *pgm* only leads to a 12 and 25% difference in cell area, respectively [5]. Apart from the most extreme acidic conditions, nearly all of the pH-dependent changes in size were restricted to changes in cell length and were independent of changes in growth rate, media composition, or buffering capacity (Fig 1C and 1D; S1 Fig, S2 Table). To independently validate that pH alters cell size homeostasis in live cells, we used time lapse imaging to measure the cell length added between divisions, a property of the adder model of cell size homeostasis [1,2]. Consistent with our findings in fixed cells, cells cultured in acidic medium added a shorter length from birth to division than cells grown at neutral and alkaline pH (Fig 1E).

pH-dependent changes in size were not unique to MG1655 or even to *E. coli*. We observed similar effects of pH on cell area in *E. coli* strain W3110, another commonly used K-12 laboratory isolate, and in the evolutionarily distant Gram-positive coccus *Staphylococcus aureus* (S3 Fig). The average volume of *S. aureus* cells was reduced by ~48% during growth in acidic (pH 5.5) medium compared to alkaline (pH 8.0) medium (average volume of $1.02 \pm 0.03$ compared to $2.10 \pm 0.17$ $\mu m^3$). Likewise, during this work two separate studies noted the size of *Streptococcus pneumoniae* and *C. crescentus* also increases during growth in alkaline medium [25,26]. Altogether, these findings establish environmental pH as a mediator of size across evolutionarily distant bacterial species.

### Acidic pH suppresses conditional mutants of late divisome proteins and bypasses the essentiality of FtsK

Our observation that pH-dependent changes in *E. coli* cell size were restricted to changes in cell length and were independent of changes in mass doubling time (Fig 1C; S2 Table) indicated divisome assembly and/or activity may be pH sensitive. In *E. coli* assembly of the 'core' cell division machinery is a sequential process [13]. First, the so-called "early" division proteins—including the cytosolic tubulin homolog FtsZ, membrane anchor ZipA, and membrane-associated actin homolog FtsA—form a dynamic, discontinuous ring-like structure at midcell [27–29]. Subsequently, a series of "late" division proteins containing transmembrane and periplasmic domains becomes enriched at the septal ring; these include the DNA translocase FtsK [30,31], the regulatory FtsQLB complex [20,21,32], and the septal cell wall synthesis transpeptidase and glycosyltransferase pair, FtsI (also known as PBP3) and FtsW [33]. In the

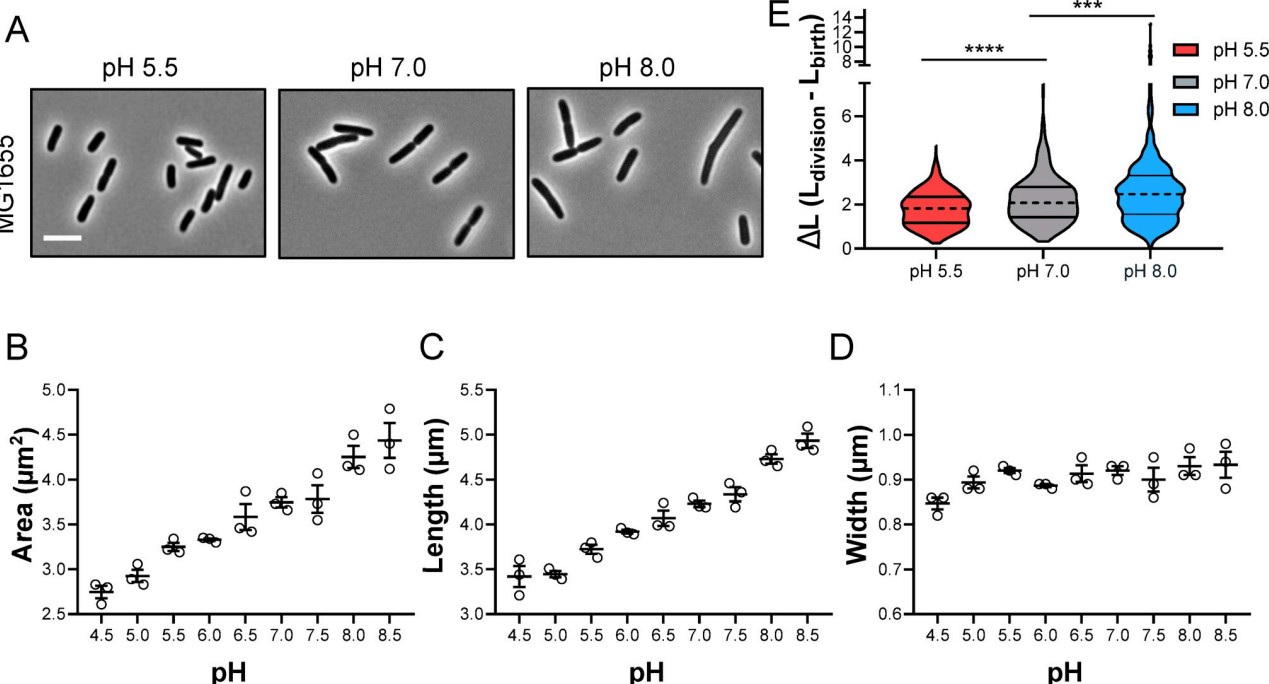

**Fig 1. Environmental pH influences *E. coli* cell size.** A) Representative micrographs of MG1655 grown to steady state in LB media + 0.2% glucose at pH 5.5, 7.0, and 8.0 and collected at $OD_{600}$ ~0.1–0.2 for imaging. Scale bar denotes 5 μm. B-D) Mean cell area (B), cell length (C), and cell width (D) for MG1655 grown in LB media + 0.2% glucose from pH 4.5–8.5. Individual points denote mean population measurement for each biological replicate. Error bars represent standard error of the mean. Significance shown in S2 Table. E) Change in length from beginning to end of the cell cycle for individual cells grown in LB media + 0.2% glucose at pH 5.5 (n = 450), pH 7.0 (n = 489), or pH 8.0 (n = 461) from two independent experiments. Dotted line represents median length added, and straight lines indicate quartiles. Significance was determined by Kruskal-Wallis test, corrected for multiple comparisons with Dunn's test.

final phase of division in *E. coli*, FtsN accumulates at midcell and is believed to "trigger" septal cell wall synthesis and constriction through interactions with the early and late divisome components [20,21,34–36]. In addition to the essential division proteins described above, there are nearly a dozen non-essential or conditionally essential factors involved in divisome stabilization (e.g., ZapBCD and FtsP) [37–40], cell wall synthesis (e.g., PBP1a and PBP1b) [41,42], cell wall hydrolysis (e.g., AmiA, AmiB, and AmiC) [43], and regulation of cell wall remodeling (e.g., FtsEX) [44,45]. The sheer number of proteins involved in division imply many possible integration points through which pH may modulate division to tune cell size. Based on our previous finding that pH impacts the activity of periplasmic cell wall enzymes [46], we speculated that division proteins with periplasmic domains would be the most likely regulatory targets of pH.

To identify the specific stage(s) of cell division influenced by pH, we took advantage of a set of heat-sensitive alleles of essential cell division genes. These conditional mutants played a historically important role in parsing the key functions of the essential components of divisome and associated modulatory proteins [47,48]. Suppression of the heat-sensitive phenotype of these conditional mutants under growth-restrictive conditions (LB- no salt, 37 or 42°C) suggests a positive influence on the division machinery while enhancement of heat sensitivity under typically growth-permissive conditions (LB, 30°C) indicates a negative influence on the division machinery. We assessed the impact of pH on the growth of a subset of heat-sensitive mutants, including alleles of both early cytoplasmic division genes [*ftsZ84* (G105S) and *ftsA27* (S195P)] and late periplasmic division genes [*ftsK44* (G80A), *ftsQ1* (E125K), and *ftsI23* (Y380D)].

Although growth of the early cytoplasmic division mutants was insensitive to pH, low pH (5.5) suppressed the heat sensitivity of the late periplasmic division mutants, and conversely, high pH (8.0) enhanced it (Fig 2A and 2B). Importantly, this effect was not allele specific. Additional heat-sensitive variants of FtsZ (*ftsZ25*), FtsA (*ftsA12*), and FtsI (*ftsI2158*) behaved similarly to their previously tested cognates, although *ftsA12* heat sensitivity was modestly enhanced at pH 8.0 (S5 Fig). When we expanded the tested pH range from pH 4.5–9.0, the heat sensitivity of the strains encoding *ftsK44*, *ftsQ1*, and *ftsI23* was consistently suppressed between pH 5.0 and pH 6.5 and enhanced between pH 7.5 and pH 9.0 (S5 Fig). Notably, these pH ranges correspond to conditions in which the wild type cells have decreased and increased average cell lengths, respectively (Fig 1C). We did not observe changes in heat sensitivity in strains encoding *ftsZ84* or *ftsA27* in the expanded pH range (S5 Fig). We also ruled out the contribution of the accessory periplasmic divisome proteins FtsP, PBP1a, and PBP1b, which had been previously shown to be stress or pH responsive [40,46]; cells defective in each of the aforementioned proteins still exhibited pH-dependent changes in cell size (S4 Fig).

The complete suppression of heat sensitivity in *ftsK44* and *ftsI23* mutants at very high temperature (42˚C) suggested these genes may be dispensable for divisome activity in acidic

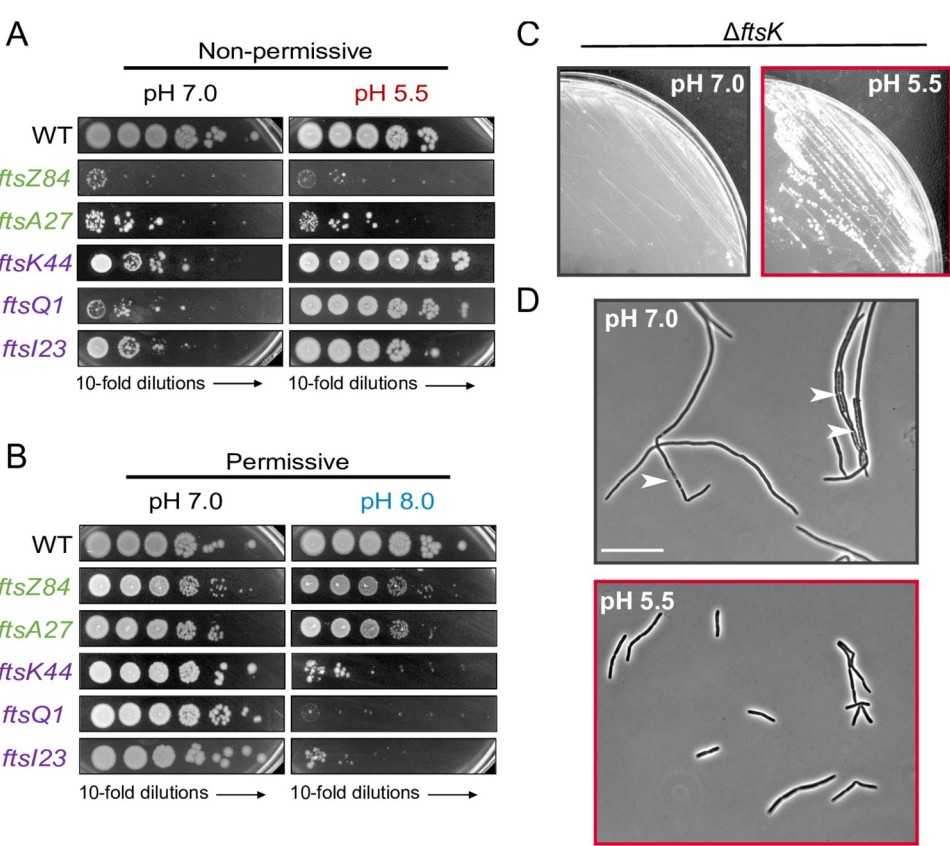

**Fig 2. Acidic pH stabilizes late division proteins and bypasses the essentiality of FtsK.** A-B) Representative plating efficiency for strains producing heat-sensitive variants of early division proteins (PAL2452, *ftsZ84*; WM4107, *ftsA27*) and late division proteins (WM2101, *ftsK44*; EC3433, *ftsQ1*; WM4649, *ftsI23*) when grown under non-permissive conditions (A) or permissive conditions (B) as a function of agar plate pH. Image is representative of three biological replicates. C) Comparison of growth of MG1655 Δ*ftsK::kan* strain (EAM1311) cultured on LB agarose plate at neutral (left) or acidic pH (right) at 30 ˚C. D) Comparison of cell morphology of MG1655 Δ*ftsK::kan* strain (EAM1311) grown for 2 hours in LB liquid media at neutral (top) or acidic pH (bottom). Arrowheads indicate lysed cells. Scale bar denotes 20 μm.

media. To test this, we attempted to transduce deletion alleles of each gene into wild type cells under acidic (pH 5.5) and neutral conditions at 30˚C. Although we were unable to delete the native *ftsI* even when we produced a catalytically inactive variant of FtsI (S307A) from a plasmid [49], we were able generate stable transductants with the *ftsK::kan* allele when the cells were grown in acidic media (Fig 2C). FtsK null mutants were slightly elongated when cultured in acidic media but rapidly filamented and lysed when transferred to neutral pH (Fig 2D). In total, these findings are consistent with acidic pH stabilizing one or more late division proteins, and this is sufficient to bypass the essential activity of FtsK.

## Septal recruitment of the terminal division protein FtsN is pH sensitive

To directly visualize the effect of pH on the assembly of the division machinery and narrow down which phase(s) of division may be pH responsive, we imaged midcell recruitment of a subset of GFP-tagged division proteins. We selected fusion proteins that spanned the divisome assembly hierarchy, including early cytoplasmic proteins FtsZ and FtsA, late periplasmic proteins FtsL and FtsI, and the terminal periplasmic division protein FtsN (Fig 3A). All constructs were integrated at the lambda locus in otherwise wild type MG1655 cells with the gene of interest expressed from an IPTG-inducible promoter. Production of these fusion proteins leads to a characteristic midcell ring of fluorescence for a fraction of the cell cycle proportional to their lifetime at the septum [50]. Because populations of *E. coli* cells are unsynchronized and thus all cell cycle stages are represented at a single time point, comparison of septal ring frequency (i.e., the percentage of cells that exhibit midcell localization of the protein of interest) across conditions can be used as a proxy for changes in assembly dynamics and/or enrichment of division proteins at midcell [50]. IPTG levels were titrated such that fusion protein production did not disrupt pH-dependent changes in size (S6 Fig).

When we compared the septal ring frequency of the fusion proteins at pH 5.5, 7.0, and 8.0, cells producing GFP-FtsN exhibited a striking pH-dependent difference in midcell localization of the protein (Fig 3B–3D; S3 Table). GFP-FtsN was significantly enriched at midcell in acidic media (~30% septal localization frequency) and reduced in alkaline media (~15% septal localization frequency). This trend held across an expanded pH range (4.5–8.5) and was inversely proportional to changes in cell length (Fig 3E). To validate the septal ring frequency analysis, we quantified midcell GFP-FtsN intensity across pH conditions using the ImageJ plugin Coli Inspector [50]. This analysis confirmed midcell enrichment of FtsN at a reduced cell size in acidic media (S7 Fig). This analysis also revealed a modest increase in GFP-FtsI intensity (S7 Fig); however, subsequent assessment of GFP-FtsI's septal localization frequency across a wider pH range did not uncover a consistent correlation between pH and ring frequency, as we observed for GFP-FtsN (Fig 3D; SI Appendix, S8 Fig). Consequently, we elected to focus on FtsN's contribution to pH-dependent changes in size in the present investigation.

Two models explain pH-dependent changes in midcell localization of FtsN: 1) increased expression, production, or stability of FtsN in acidic conditions and/or 2) changes in FtsN's affinity for the cytokinetic machinery. To address the former possibilities, we compared bulk levels of FtsN from cells grown in different pH environments. Neither the levels of the native or GFP-FtsN fusion protein varied as a function of pH by immunoblot (Fig 3F; SI Appendix, S9 and S10 Figs). We did observe the appearance of an FtsN degradation or processed product, but its abundance was not predictably correlated with pH (SI Appendix, S9 Fig). Altogether, these data indicate observed changes in ring frequency are likely due to an increase in affinity for the septal ring under acidic conditions.

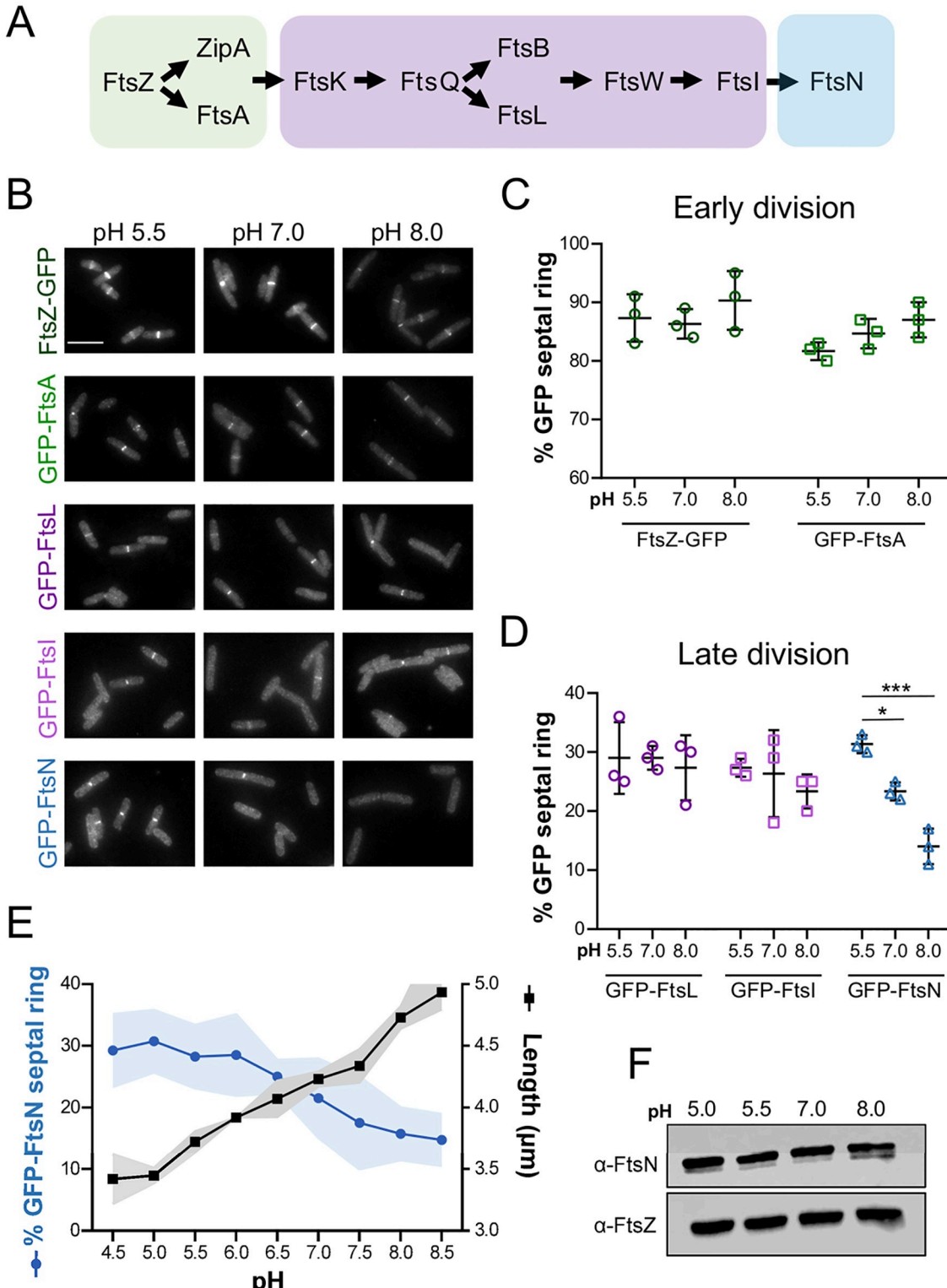

**Fig 3. Septal recruitment of the terminal division protein FtsN is pH sensitive.** A) Schematic depicting recruitment hierarchy of the early division proteins (green), late division proteins (purple), and final division protein FtsN (blue) to midcell. B) Representative micrographs of MG1655 derivatives producing the indicated GFP-tagged division proteins (BH330, FtsZ-GFP; EAM410, GFP-FtsA; PAL3700, GFP-FtsL; EAM412, GFP-FtsI; EAM621, GFP-FtsN). Cells were cultured to steady state in LB media at pH 5.5, 7.0, and 8.0 and collected for imaging at $OD_{600} \sim 0.1$–0.2. Scale bar denotes 5 μm. C-D) Mean percentage of cells that score positive for a GFP

septal ring of the indicated early (C) and late (D) GFP-tagged division proteins in LB media at pH 5.5, 7.0, and 8.0. Individual points depict population mean of individual biological replicates. Error bars represent standard deviation. Significance was determined using a two-way ANOVA, corrected for multiple comparisons with Sidak's test. E) Comparison of mean GFP-FtsN septal ring frequency (EAM621) and mean cell length (MG1655) from pH 4.5–8.5. Cell length data is from Fig 1C. Shaded region denotes the error of the measurement (SD for ring frequency; SEM for cell length). F) Representative immunoblot for FtsN and FtsZ levels in MG1655 Δ*malE::kan* (CW142) cultured to steady state in LB media at pH 5.0, 5.5, 7.0, and 8.0. Quantification shown in SI Appendix, S9 Fig.

## Enrichment of septal FtsN in acidic media does not require its glycan binding domain

We next sought to determine the regions of FtsN required for pH-dependent recruitment to midcell. FtsN is a bitopic inner membrane protein with no known enzymatic activity. It possesses three regions with characterized roles in cell division: 1) a short, N-terminal cytoplasmic patch that directly interacts with the early division protein FtsA [51,52]; 2) an alpha helical region in the periplasm referred to as the constriction control domain (CCD, amino acids 75–93) that is believed to activate the septal cell wall synthesis machinery [20,35,52]; and 3) a periplasmic C-terminal SPOR domain that binds denuded cell wall glycans produced upon constriction initiation [35,53] (Fig 4A). While the SPOR domain is the primary septal localization determinant of FtsN, the FtsA interaction interface and CCD are believed to a play a role in the initial recruitment of FtsN to the septum, at least prior to the onset of constriction [52–54]. The CCD is the only region of FtsN strictly essential for viability [20,35].

To clarify which, if any, of these interactions are required for pH-sensitize recruitment of FtsN to midcell, we compared the septal localization frequency of truncations containing both the FtsA interaction interface and CCD (1–243 and 1–105) or only the FtsA interaction

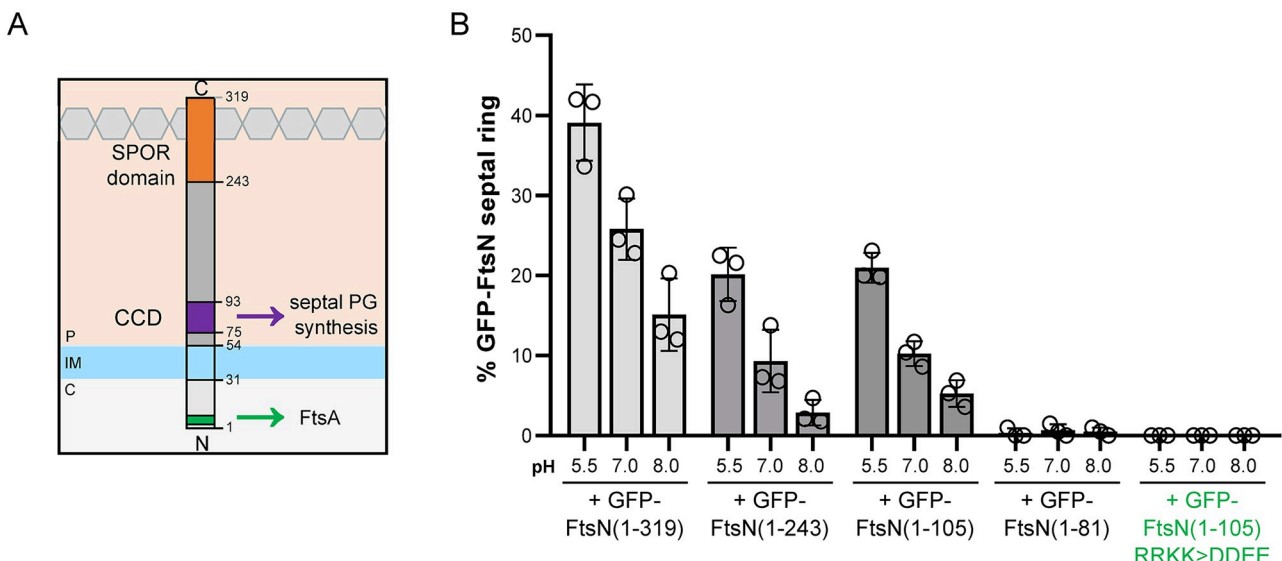

**Fig 4. pH-dependent recruitment of FtsN to midcell does not require the SPOR domain.** A) Schematic depicting the major features of FtsN, including an FtsA interaction interface (green), essential constriction control domain (CCD, purple), and peptidoglycan-binding SPOR domain (orange). B) Mean percentage cells that score positive for a GFP-FtsN septal ring when producing either full length GFP-FtsN (pCH201), GFP-FtsN(1–243) (pCH354), GFP-FtsN(1–105) (pMG12), GFP-FtsN(1–81) (pMG13), or GFP-FtsN(1–105) RRKK>DDEE (pMG12-RRKK>DDEE) from a plasmid in the wild type background (MG1655) Cells were grown to steady state in LB media with 25 μM IPTG and collected for imaging at OD₆₀₀ ~ 0.1–0.2. Individual points depict population mean of individual biological replicates. Error bars represent standard deviation.

interface (1–81) fused to GFP at the N-terminus and expressed from a plasmid at low induction levels. We could not visualize the SPOR domain alone, as periplasmic GFP fusions exhibit pH-dependent changes in brightness even after fixation [55]. If any of these regions affects FtsN septal recruitment across pH environments, we expect to observe an increase in septal localization in acidic conditions, similar to what we previously observed for the full length GFP-FtsN (Figs 3 and 4B).

Surprisingly, although loss of the SPOR domain decreased the overall percentage of cells scoring positive for a GFP-FtsN ring, it was not required for pH-dependent differences in septal localization frequency. Truncations containing just the FtsA interaction domain and the CCD—GFP-FtsN(1–243) and GFP-FtsN(1–105)—remained pH sensitive (Fig 4B). In contrast, cells producing GFP-FtsN(1–81) rarely were scored positive for a septal ring under any pH condition, similar to what had been previously observed [35]. To interrogate the contribution of the FtsA interface, we compared the septal ring frequency of a GFP-FtsN(1–105) variant in which a patch of conserved basic residues in the N-terminus was mutated to acidic amino acids (residues 16–19, RRKK>DDEE). This charge swap has been shown to impair the FtsN-FtsA interaction both *in vitro* [56] and *in vivo* [52]. Consistent with a role for FtsA interaction in septal recruitment, this variant failed to localize to midcell under any pH condition (Fig 4B). Overall, our data indicate that the CCD and FtsA interaction interface—but not the SPOR domain—are required for differential recruitment of FtsN to the septum across pH environments.

## Overexpression of *ftsN* decreases cell length in rich media

If FtsN localization to the otherwise mature divisome is sufficient to trigger constriction, increasing FtsN's likelihood of interaction with other components of the division machinery via overexpression should lead to reductions in cell length. To test this model, we overexpressed *gfp-ftsN* from a plasmid in the wild type background at neutral pH. Consistent with FtsN serving as a division "trigger" and with previous work in *C. crescentus* [19], we observed an induction-dependent decrease in cell length of up to ~15%, which correlated with an increase in septal localization frequency (Fig 5A, 5B and 5D; S3 Table). Cell width did not decrease upon *ftsN* overexpression and in fact, modestly increased (Fig 5C), possibly reflecting a competition between the cell division and cell elongation machineries for a shared pool of precursors [28]. These results conflict with some studies reporting toxicity and a modest increase in cell length with *ftsN* overexpression [13,48,57]. To confirm our findings, we repeated this experiment using a separate untagged expression construct and again observed a similar reduction in cell length (S12 Fig). Differences in growth conditions likely explain at least part of the discrepancy between *ftsN* overexpression phenotypes [13,19,48,57]. While overproduction of FtsN in rich medium reduces size and does not impact mass doubling time (Fig 5; S12 Fig and S4 Table), we find overproduction of FtsN in minimal medium (AB + 0.2% glycerol) results in a severe growth defect (S12 Fig).

To identify the regions of FtsN that are sufficient to reduce length in rich media, we overexpressed a series of *ftsN* truncation mutants in wild type cells at neutral pH and measured their size [35]. Our data demonstrate the N-terminal 105 amino acids of FtsN, which include both the FtsA interaction interface and the CCD, are sufficient for overexpression-dependent reductions in cell length (Fig 5E). Overexpression of *ftsN* encoding only part of the CCD (1–81, 1–90), the CCD alone (71–105 targeted to the periplasm with a TorA signal peptide), or the SPOR domain alone (243–319 targeted to the periplasm with a TorA signal peptide) did not significantly reduce length despite being produced at similar levels (S13 Fig). Somewhat surprisingly, the SPOR domain was not required for reductions in length, suggesting a direct interaction with the

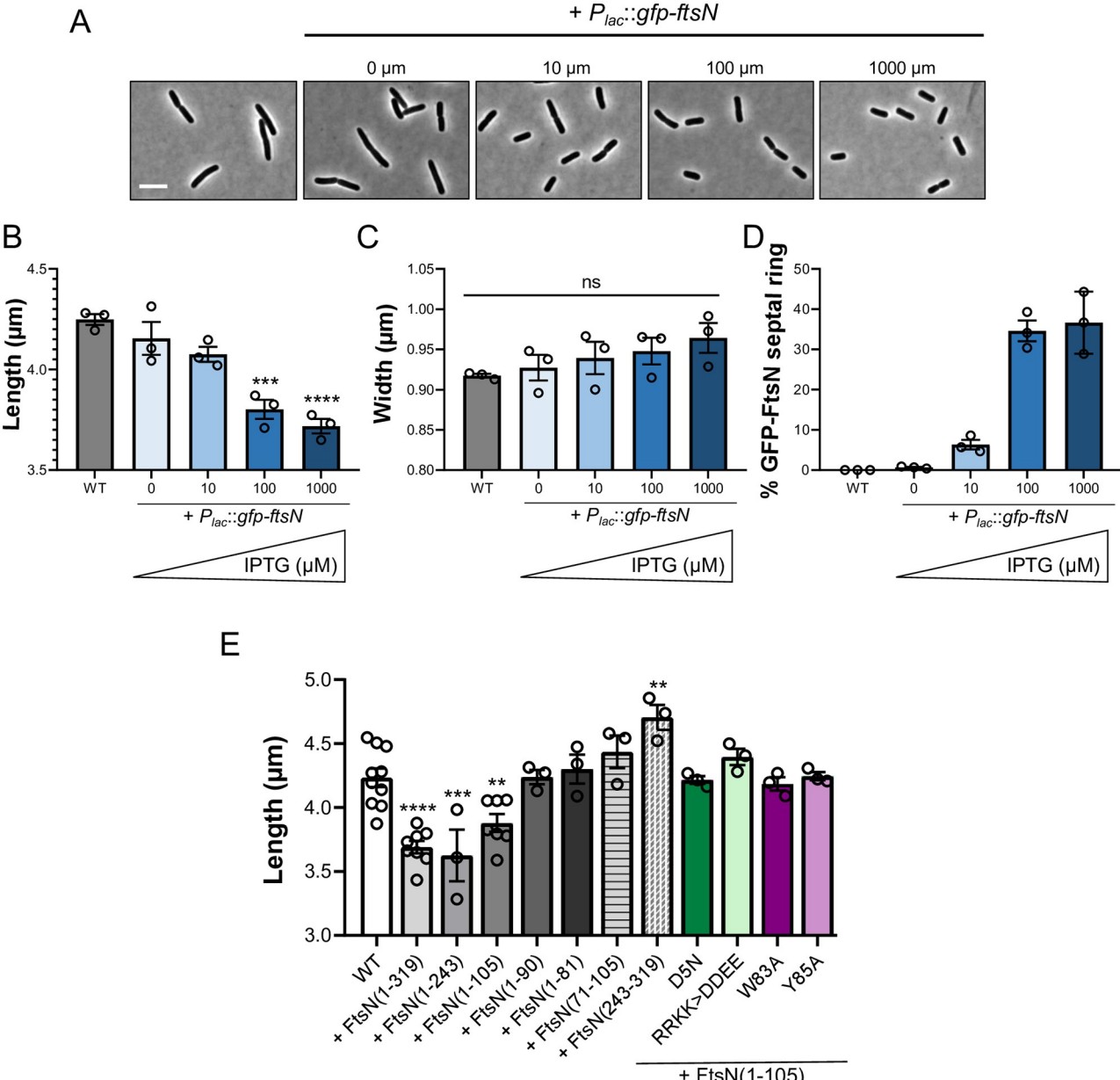

**Fig 5. Overexpression of *ftsN* reduces cell length.** A) Representative micrographs of MG1655 overexpressing *gfp-ftsN* (MG1655/pCH201) grown to steady state with varying levels of inducer (IPTG) and collected for imaging at $OD_{600}$ ~0.1–0.2. Scale bar denotes 5 μm. B-D) Mean cell length (B), cell width (C), and GFP-FtsN septal ring frequency (D) of cells overexpressing *gfp-ftsN* grown to steady state LB media (MG1655/pCH201) with varying levels of inducer (IPTG). Individual points depict population mean from each biological replicate. Error bars represent standard error of the mean (B, C) or standard deviation (D). Significance was determined by a one-way ANOVA, normalized for multiple comparisons with Dunnett's test. E) Mean cell length of MG1655 overexpressing the indicated *gfp-ftsN* truncations and point mutants from a plasmid during growth in LB media. All strains harboring a construct were induced with 1 mM IPTG. Individual points depict population mean from each biological replicate. Error bars represent standard error of the mean. Significance was determined by a one-way ANOVA, normalized for multiple comparisons with Dunnett's test.

peptidoglycan is not necessary for FtsN's role in size control [53]. Overexpression of *ftsN(1–105)* mutants impaired for FtsA interaction (D5N, RRKK>DDEE) [51,52,56] or essential CCD activity (Y83A, W85A) [20] failed to reduce length despite being stably produced (Fig 5E and 5F; S13 Fig). Collectively, this functional analysis establishes the FtsA interaction interface and the CCD as requirements for FtsN-mediated reductions in cell length.

## The size of cells expressing gain-of-function mutants *ftsA** and *ftsL** is insensitive to pH

Gain-of-function alleles of *ftsA*, *ftsL*, *ftsB* and *ftsW* have been identified that mimic the stimulatory effect of FtsN on the divisome and consequently bypass FtsN's essential function in *E. coli*. Cells expressing these mutants are constitutively short, independent of changes in growth rate [18,20,21,45,58]. We reasoned that if environmental pH modulates division through a related mechanism, the size of cells expressing hypermorphic alleles would be insensitive to pH (Fig 6A). To test this model, we compared size and FtsN localization in two of most well-studied division hypermorphs, *ftsA** (R286W) and *ftsL** (E88K), at pH 5.5, 7.0, and 8.0. The size of *ftsA** and *ftsL** mutants was invariant across pH conditions (Fig 6B). At the same time, the frequency of septal GFP-FtsN was higher in the hypermorphic strains (41% and 38%, respectively, compared to 21% of wild type cells) independent of differences in GFP-FtsN protein levels (Fig 6C; S12 Fig). GFP-FtsN septal localization remained high across all tested pH conditions (S3 Table).

These data are consistent with two models for pH-mediated division activation: 1) growth in acidic media recruits FtsN to the septum more efficiently, causing activation of the divisome at a reduced cell size, or 2) pH influences the divisome activation state independent of FtsN. In the latter model, enhanced FtsN recruitment under acidic conditions may be a consequence, rather than the cause, of pH-dependent activation of one or more upstream division proteins, possibly due to the proposed self-reinforcing nature of the divisome [59]. To differentiate between these models, we attempted to deplete FtsN in wild type cells in acidic media. If acidic pH activates the divisome independent of FtsN, we anticipated that less FtsN would be required to sustain growth in acidic media, as is seen for the hypermorph mutants. Consistent with previous work [20,36,60], *ftsA** and *ftsL** mutants tolerated significant depletion of FtsN irrespective of pH environment. However, we were unable to deplete FtsN in wild type cells in any pH condition tested (Fig D; S14 Fig). This result indicates FtsN is required for low pH-mediated division activation, favoring model in which acidic pH activates division either through FtsN alone or FtsN together with upstream divisome proteins (Fig 6A).

## Discussion

While metabolic control of cell size has been a topic of investigation for nearly sixty years [61], comparatively little is known about the impact of the physical and chemical environment on cell size. Here, we make the surprisingly observation that *E. coli* cell division and cell size are remarkably sensitive to environmental pH, a property that varies widely across the niches this organism inhabits in the wild [23,24]. Specifically, we find that growth in acidic media stimulates cytokinesis in *E. coli* at a smaller volume than that of cells grown in neutral media; conversely, alkaline conditions increase the size at division (Fig 1). The differences in average length between cells cultured at pH 4.5 and pH 8.5 exceeds 40%, over three times the contribution of metabolic regulator OpgH to cell size [5]. Significantly, pH impacts cell length independent of changes in mass doubling time (S2 Table), contributing to a growing body of evidence suggesting size is not necessarily coupled to growth rate [12,62,63].

### FtsN as an integration point for pH-dependent changes in cell size

Our data indicate acid-dependent division activation is governed at least in part by increases in the affinity of the *E. coli* terminal division protein and so-called division "trigger," FtsN, for other periplasmic components of the cytokinetic ring. An alkaline environment likely has the opposite effect—reducing affinity between divisome components and inhibiting FtsN

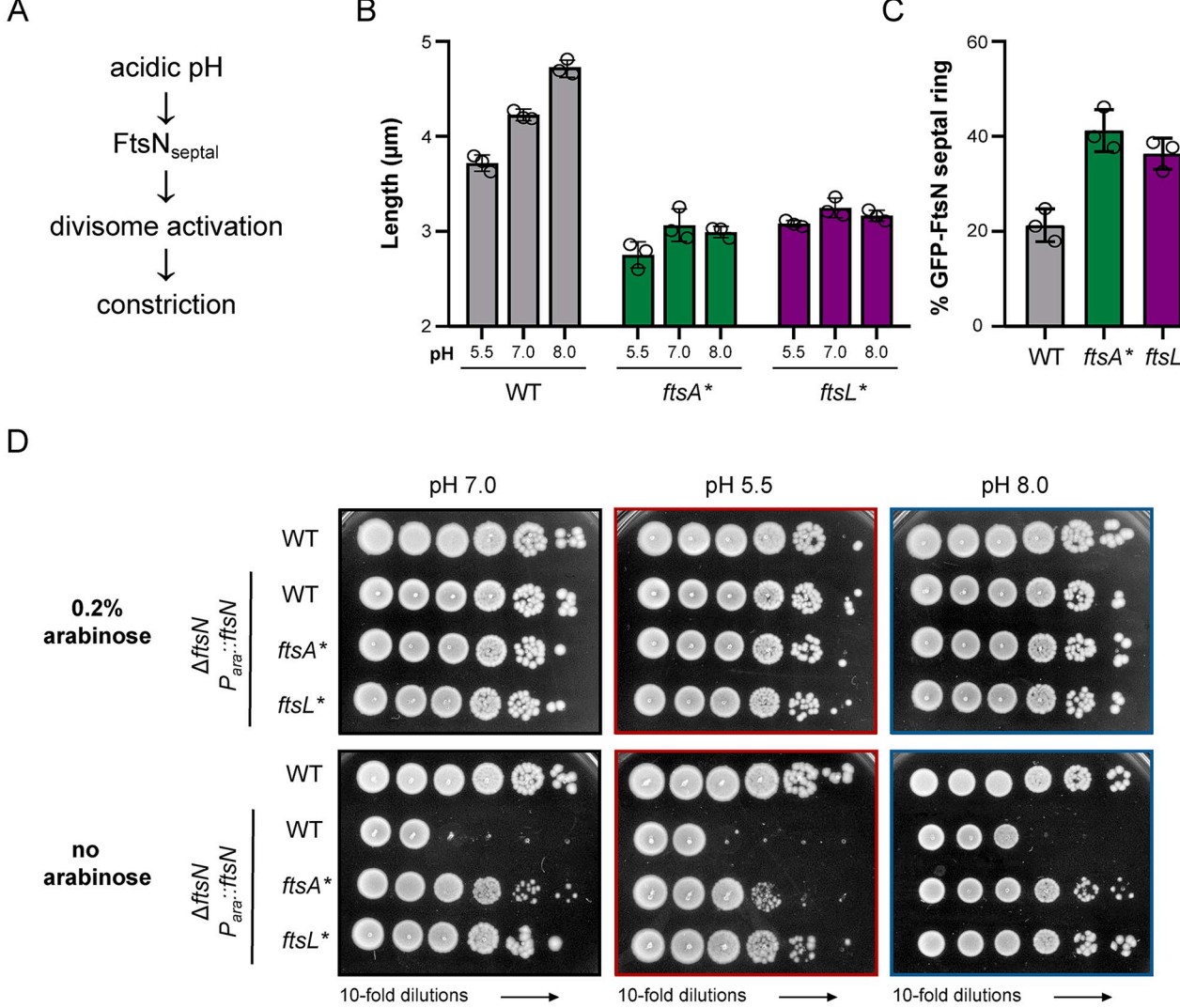

**Fig 6. $ftsA^*$ and $ftsL^*$ gain-of-function mutants are insensitive to pH.** A) Genetic model for pH-dependent reductions in cell length. B) Mean cell length of $ftsA^*$ (BH142) and $ftsL^*$ (MT13) grown to steady state LB media + 0.2% glucose at pH 5.5, 7.0, and 8.0 and collected at $OD_{600} \sim 0.1$–0.2 for imaging. MG1655 data from Fig 1C is shown for comparison. Individual points denote population mean for each biological replicate. Error bars represent standard error of the mean. Significance was determined using a two-way ANOVA, corrected for multiple comparisons with Sidak's test. C) Mean GFP-FtsN septal ring frequency for $ftsA^*$ (EAM747) and $ftsL^*$ (EAM749) cultured to steady state in LB media at pH 5.5, 7.0, and 8.0 and collected at $OD_{600} \sim 0.1$–0.2 for imaging. EAM621 (MG1655 GFP-FtsN) data Fig 3D is shown for comparison. D) Representative plating efficiency for $ftsN$ depletion in WT (HSC074/pBAD33-ftsN), $ftsA^*$ (EAM719/pBAD33-ftsN), and $ftsL^*$ (EAM723/pBAD33-ftsN) cells at pH 5.5 (middle), 7.0 (left), and 8.0 (right). Image is representative of three biological replicates.

recruitment (Fig 7). Several independent lines of inquiry support this conclusion. First, we find that FtsN septal accumulation is inversely proportional to pH-dependent changes in cell length. During growth in acidic conditions, cells are short, and septal FtsN is abundant. Conversely, during growth in alkaline media, cells are long, and septal FtsN is depleted (Fig 3). Secondly, consistent with changes in midcell FtsN accumulation driving pH-dependent differences in cell length, overexpression of FtsN is sufficient to promote cytokinesis at a reduced cell volume, at least during growth in rich media (Fig 5; S12 Fig). Importantly, overexpression of other late division proteins has not been associated with reductions in cell size. Simultaneous overexpression of the $ftsQ$, $ftsL$, and $ftsB$ causes cell filamentation [64], and

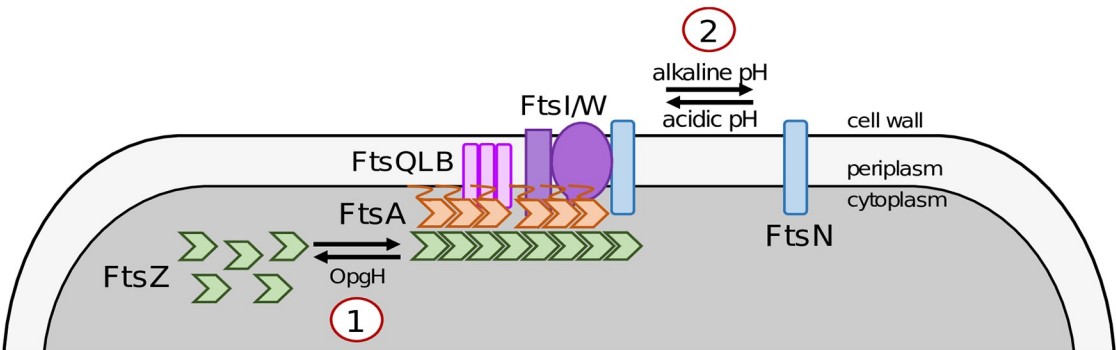

**Fig 7. Simplified model of known environmental regulators of cell division and cell size in *Escherichia coli*.** 1) Growth in carbon-rich media leads to intracellular accumulation of the metabolite uridine disphosphate (UDP)-glucose. UDP-glucose activates moonlighting glucosyltransferase OpgH, which antagonizes FtsZ assembly and leads to an increase in cell length. 2) Environmental pH alters in the affinity for FtsN for the midcell. Growth in acidic medium enhances recruitment of FtsN to midcell, reducing cell length. Conversely, growth in alkaline medium inhibits FtsN accumulation at the midcell, increasing cell length.

overexpression of *ftsI* also modestly increases cell length (S12 Fig). Finally, in further support of a direct role for FtsN in low pH-mediated division activation, FtsN cannot be depleted in acidic media (Fig 6D), which would be expected if division activation occurred in an FtsN-independent manner [20,36,60].

Several additional, less direct pieces of evidence also implicate FtsN's involvement in pH-dependent divisome activation. Overexpression of FtsN suppresses the heat sensitivity of cells encoding variants of FtsA, FtsK, FtsQ, and FtsI and bypasses the essentiality of FtsK (S11 Fig) [48,65,66], similar to the phenotypes we observe when culturing the mutants in acidic media (Fig 2). We also observed an increase in cell chaining in alkaline pH, particularly during growth in MOPS minimal media (S1 Fig). This latter observation is consistent with FtsN's role in recruiting the septal amidases, which are required for efficient daughter cell separation following constriction [43,67].

FtsN is an attractive integration point for cell size control. As the final essential division protein enriched at the septum [34,68], FtsN has long been believed to "trigger" cytokinesis, and its presence at the septum is correlated with visible constriction [35,69,70]. Intriguingly, FtsN interacts with and regulates both early cytoplasmic and late periplasmic divisome proteins [20,21,71,72], potentially allowing it coordinate the activities of disparate components of the cell division apparatus. Indeed, recent work suggests FtsN may promote cytokinesis through two mechanisms: 1) stabilizing FtsZ filaments through its FtsA interaction interface [56], and 2) activating the septal cell wall synthesis enzymes FtsI and FtsW in the periplasm through its essential constriction control domain (CCD) [20,73]. Our data reinforce contribution of both the FtsA interaction interface and the CCD in FtsN's ability to promote cytokinesis. FtsN mutants defective for either domain fail to localize to the cytokinetic ring or to reduce size when overexpressed (Figs 4 and 5). Apart from promoting cytokinesis, FtsN also activates PBP1b [64,74,75], a nonessential cell wall synthesis enzyme speculated to play a role in cell wall repair during normal growth of the *E. coli* peptidoglycan sacculus [46,76–78]. Midcell enrichment of FtsN in acidic media may also augment PBP1b activity at the septum and thus direct it to the region with the highest rates of peptidoglycan synthesis. In support of this model, PBP1b defective cells form septal bulges and lyse upon exposure to extremely acidic (pH < 4.8) conditions [46].

Our data suggest a model for pH-mediated division activation in *E. coli*. Acidic growth conditions enhance recruitment of FtsN to the septum through an as of yet unknown mechanism.

On the cytoplasmic side of the inner membrane, the FtsN-FtsA interaction stabilizes treadmilling FtsZ filaments, possibly by modulating FtsA turnover and/or polymeric state [56,59]. In the periplasm, the CCD of FtsN converts the septal cell wall synthesis enzymes into an active state either through direct interaction with septal cell wall synthases FtsI and/or FtsW or indirectly through the regulatory FtsQLB complex [20,21,64,73]. Cytoplasmic and periplasmic interactions initiate a positive feedback loop, promoting FtsN accumulation iteratively until a threshold level of divisome activity is reached to promote constriction. Constriction exposes denuded glycans in the septal peptidoglycan through the action of the amidases and further enhances FtsN recruitment via the SPOR domain [53]. The iterative, self-reinforcing nature of this activation cascade ensures completion of cross-wall synthesis and the viability of the two daughter cells. Under alkaline conditions, impaired FtsN recruitment delays initiation of constriction until a larger cell size is reached.

While our data favor a specific role for FtsN in driving pH-dependent changes in *E. coli* size, we anticipate that pH has pleiotropic effects on the division machinery. In particular, we speculate that pH affects many, if not all, of the extracellular divisome proteins either by directly impacting activity or through more subtle changes in protein-protein interactions. Consistent with this model, we and others have previously identified a handful of pH sensitive extracellular cell wall enzymes with diverse enzymatic functions [46,79–81]. *S. aureus* and *S. pneumoniae* still undergo pH-dependent changes in size but lack identifiable homologs of FtsN (S3 Fig) [25], indicating the existence of additional pH-responsive divisome components at least in these organisms. Recent technological advancements, including the use of FRET biosensors to probe interactions between division proteins and new methods to assay activity and interactions between the membrane-associated divisome components, offer promising avenues to dissect the impact of pH specific divisome interactions in future studies [64,82].

## A threshold level of divisome activity dictates cell size

Our data support a refinement of the threshold model for cytokinesis. As first proposed by the Jun lab, the current threshold model states division is coordinated with cell size via growth rate-dependent accumulation of key division proteins to threshold numbers at the future site of septation [3]. In light of our findings, we favor a "general threshold" model, in which a threshold level of divisome activity must be attained prior to cytokinesis and thus dictates homeostatic cell size. In this revised model, cell division can be coordinated with cell size through disparate, if complementary, mechanisms. These mechanisms include changes in the specific number of critical divisome proteins at midcell as proposed by Si *et al.* and others [3–5,19,83,84], alterations in the affinity of divisome proteins for the cytokinetic ring (this work), and changes in the activation state of key regulatory proteins within the ring itself [20,21,85]. Significantly, tuning divisome activity through a variety of mechanisms increases flexibility and allows cells to modulate size in response to a variety of signals. Cytosolic signals (e.g., metabolic state, DNA replication status) may be communicated to the division machinery via regulation of FtsZ or other cytosolic divisome components [4,5], whereas environmental signals that preferentially affect the properties of the periplasm (e.g., pH, osmolarity) could be relayed through differential assembly or activation of the late division machinery rather than through canonical signal transduction cascades (Fig 7).

## The physical and chemical environment as a mediator of extracellular processes

More broadly, our results point to the chemical and physical environment as an important mediator of extracytoplasmic processes [86]. Environmental pH, in particular, appears to

modulate the activity of several cell wall enzymes in *E. coli*, including the class A PBPs [46], the lytic transglycosylase MltA [79], and the carboxypeptidases PBP5 and PBP6b [80]. The septal cell wall synthesis machinery in *Salmonella* is similarly pH sensitive [81]. However, pH sensitivity may represent just the 'tip of the iceberg'. Although the underlying molecular mechanisms remain unclear, media osmolarity affects the growth of cells harboring conditional mutants of division genes, dictates essentiality of FtsEX for division, and modulates cell size in *E. coli* [47,87,88]. Extracellular metal availability also alters the activity of several nonessential cell wall enzymes [89–91]. Improved understanding of how extracellular processes cope with dynamic environments promises to shed light on how single celled organisms survive—and thrive—across a wide range of ecological niches.

## Materials and methods

### Bacterial strains, plasmids, and growth conditions

Unless otherwise indicated, all chemicals, media components, and antibiotics were purchased from Sigma Aldrich (St. Louis, MO). Bacterial strains and plasmids used in this study are listed in S1 Table, respectively. All *E. coli* experiments, with the exception of Fig 6D and SI Appendix S3 Fig, were performed in the MG1655 background, referred to as 'wild type' in the text. P1 transduction was used to move alleles of interest between strains, and transductants were confirmed with diagnostic PCR. Mutants were generated using the Q5 Site-Directed Mutagenesis Kit (New England Biolabs) and confirmed with sequencing. Unless otherwise indicated, *E. coli* strains were grown in lysogeny broth (LB) media (1% tryptone, 1% NaCl, 0.5% yeast extract) with the pH fixed with concentrated NaOH or HCl prior to autoclaving and supplemented with 0.2% glucose. Media pH was confirmed after sterilization. *S. aureus* strains were grown in tryptic soy broth (TSB) with the pH fixed with concentrated NaOH or HCl prior to autoclaving. Where indicated, media was supplemented with 100 mM MES (pH 5.0) or HEPES (pH 7.0 or 8.0) buffers. Cells were cultured in the indicated media at 37˚C shaking at 200 rpm. When selection was necessary, cultures were supplemented with 50 μg/mL kanamycin (Kan), 30 μg/mL chloramphenicol (Cm), 12.5 μg/mL tetracycline (Tet), and/or 25–100 μg/mL ampicillin (Amp).

### Image acquisition

Phase contrast and fluorescence images of fixed cells were acquired from samples on 1% agarose/PBS pads with an Olympus BX51 microscope equipped with a 100X Plan N (N.A. = 1.25) Ph3 objective (Olympus), X-Cite 120 LED light source (Lumen Dynamics), and an OrcaERG CCD camera (Hammamatsu Photonics) or a Nikon TiE inverted microscope equipped with a 100X Plan N (N.A. = 1.25) objective (Nikon), SOLA SE Light Engine (Lumencor), heated control chamber (OKO Labs), and ORCA-Flash4.0 sCMOS camera (Hammamatsu Photonics). Filter sets for fluorescence were purchased from Chroma Technology Corporation. Nikon Elements software (Nikon Instruments) was used for image capture.

### Cell size analysis

To achieve balanced growth, cells were cultured from a single colony and grown to exponential phase (OD$_{600}$ ~ 0.2–0.6). Cultures were then back-diluted into fresh media to an OD$_{600}$ = 0.005 and grown to early exponential phase (OD$_{600}$ between 0.1–0.2) prior to being sampled and fixed for analysis. Cells (500 uL) were fixed by adding 20 μL of 1M NaPO4 (pH 7.4) and 100 μL of fixative (16% paraformaldehyde and 8% glutaraldehyde). Samples were incubated at room temperature for 15 min then on ice for 30 min. Fixed cells were pelleted, washed three times in

1 mL 1X PBS (pH 7.4), then resuspended in GTE buffer (glucose-tris-EDTA) and stored at 4˚C. Images were acquired for analysis within 48 hr of fixation. Cell length, width, and area of *E. coli* cells were determined from phase contrast images using either the ImageJ plugin Coli-Inspector (Figs 1 and 6) [50] or the MATLAB software SuperSegger (Fig 5) [92]. For *S. aureus*, cell radii were manually measured and used to calculate cell area. Cell measurements from at least 200 cells from each of at least 3 biological replicates were used to generate single point plots and histograms. Wild type or reference controls were performed during each experiment.

## Time lapse microscopy

Wild type cells in early exponential phase ($OD_{600}$ between 0.1–0.2; 5 μl) were transferred to a 1% agarose/LB + 0.2% glucose pad at the indicated pH, allowed to dry for 10 minutes, and then imaged on a Nikon TiE inverted microscope heated to 37˚C. Phase contrast images were acquired every 2 minutes for 2 hours. ΔL (change in length) for each cell was calculated from $L_{division} - L_{birth}$ in SuperSegger [92]. Cells that existed for fewer than 2 frames or more than 20 frames or grew by less than 0.5 uM between birth and death were excluded from the analysis.

## Septal ring frequency analysis

Strains producing GFP fusions were cultured, sampled, and fixed as in the section entitled 'Cell size analysis.' Induction conditions for each strain are provided in S3 Table. Phase contrast and fluorescence images were acquired on either a Nikon TiE inverted microscope or Olympus BX51 microscope. The presence of a GFP 'ring' for each cell was determined manually: cells were considered positive for a septal ring if they contained a visible band of GFP across the width of the cells or if a single spot of GFP was visible at the midpoint of an invaginating septum. Septal ring frequencies were determined from at least 200 cells from each of at least 3 biological replicates to generate single point plots.

## Heat-sensitivity assays

Strains harboring alleles that encode for heat-sensitive variants of division proteins were grown in LB (pH 7.0) at 30˚C until mid-log phase ($OD_{600}$ ~ 0.2–0.6). Cells were pelleted, washed 1x in LB- no salt medium, and resuspend in LB-no salt media to an $OD_{600} = 1.0$. Cells were diluted in LB-no salt medium, and serial dilutions $10^{-1}$ to $10^{-6}$ were plated under permissive and non-permissive conditions for each mutant with the pH of the plate varying. Plates were incubated for 20 hours. Each experiment was performed at least three times with representative images shown. The permissive condition shown for strains harboring the *ftsZ84*, *ftsA27*, and *ftsQ1* alleles is LB-no salt plates incubated at 30˚C; the non-permissive condition shown for these strains is LB-no salt plates incubated at 37˚C. The permissive conditions shown for strains harboring the *ftsI23* and *ftsK44* alleles are LB-no salt plates incubated at 37˚C; the non-permissive condition shown for these strains is LB-no salt plates incubated at 42˚C.

## Immunoblotting

Strains were grown from a single colony in LB at the indicated pH to mid-log phase ($OD_{600}$ ~0.2–0.6), back-diluted to 0.005 in 5 mL of media and grown to an $OD_{600}$ between 0.2–0.3. For experiments measuring native FtsN levels, a MG1655 *malE::kan* strain was used to eliminate cross-reactivity with the similarly sized maltose binding protein, as antiserum was raised against a FtsN-MBP fusion protein. Samples were pelleted, re-suspended in 2x Laemmli buffer to an $OD_{600}$ ~20, and boiled for ten minutes. Samples of equivalent volumes were separated on 12% SDS-PAGE gels by standard electrophoresis and transferred to PVDF membranes. Blots

were probed with FtsN (1:5000), GFP (1:2000; Abcam), and FtsZ (1:5000) rabbit antiserum and HRP-conjugated secondary antibody (1:5000–1:10000; goat anti-rabbit). Blots were imaged on a LiCor Odyssey imager. Quantitation was determined in FIJI on background subtracted images and normalized to Ponceau staining as a total protein loading control [93].

### FtsN depletion

Strains were grown from a single colony in LB (pH 7.0) in the presence of inducer (0.2% arabinose) at the indicated pH to mid-log phase ($OD_{600}$ ~0.2–0.6). Cells were pelleted, washed 3x in LB media (no inducer), and resuspend in LB to an $OD_{600}$ = 1.0. Cells were diluted in LB pH 7.0 media, and serial dilutions $10^{-2}$ to $10^{-7}$ were plated onto plates with and without inducer (0.2% arabinose) at pH 5.5, 7.0, and 8.0. Plates were incubated for 20 hours. Each experiment was performed at least three times with representative images shown.

### Statistical analysis

A minimum of three biological replicates were performed for each experimental condition unless otherwise indicated. Data are expressed as means ± standard deviation (SD) or standard error of the mean (SE). Statistical tests employed are indicated in the text and corresponding Fig legend. Analysis was performed in GraphPad Prism. No statistical methods were used to predetermine sample size. Asterisks indicate significance as follows: *, $p<0.05$; **, $p<0.01$; ***, $p<0.001$; ****, $p<0.0001$.

### Supporting information

**S1 Fig. pH-dependent changes in cell size are independent of growth medium and buffering capacity.** A-C) Cell area distribution of MG1655 grown to steady state in AB minimal medium + 0.2% glucose (A), MOPS minimal medium + 0.2% glucose (B), or LB medium supplemented with 100 mM MES (pH 5.5) or HEPES (pH 7.0 or pH 8.0) (C) and collected for imaging at $OD_{600}$ ~ 0.1–0.2. D) Fraction of cells present in chains as a function of medium pH during growth in MOPS minimal medium + 0.2% glucose. E) Change in pH as a function of optical density in unbuffered LB medium. Cells were inoculated at an $OD_{600}$ = 0.005.
(TIF)

**S2 Fig. Distribution of MG1655 cell lengths as a function of pH (A) or upon *gfp-ftsN* overexpression from pCH201 plasmid (B).** Related to Figs 1 and 5.
(TIF)

**S3 Fig. Evolutionarily distant bacteria undergo pH-dependent changes in cell size.** A-B) Representative micrographs and cell area distributions for *E. coli* strain W3110 grown to steady state in LB + 0.2% glucose (A) and *S. aureus* strain Newman grown in TSB (B) at pH 5.5, 7.0, and 8.0 and collected for imaging at $OD_{600}$ ~ 0.1–0.2. Scale bar denotes 5 μm.
(TIF)

**S4 Fig. Accessory divisome factors do not participate in pH-dependent changes in cell size.** A-B) Cell area distributions for MG1655 strains defective for PBP1a (*mrcA*::*frt*, EAM899) and PBP1b (*mrcB*::*frt*, EAM696) production during steady state growth in LB + 0.2% glucose at pH 5.5, 7.0, and 8.0. Cells were collected for imaging at $OD_{600}$ ~ 0.1–0.2. C) Representative micrographs of MG1655 strain defective for FtsP (*ftsP*::*kan*, EAM1081) during steady state growth in LB + 0.2% glucose at pH 5.5 (left) and pH 8.0 (right). Cells were collected for imaging at $OD_{600}$ ~ 0.1–0.2.
(TIF)

**S5 Fig. Mutants producing heat-sensitive variants of late division proteins are suppressed in acidic conditions and enhanced in alkaline conditions.** A) Representative plating efficiency for cells producing unique heat-sensitive variants of FtsZ (PAL2452, PAM161), FtsA (WM4107, MM61), and FtsI (WM4649, AX655) during growth at permissive (left) or non-permissive (right) conditions. B) Representative plating efficiency for cells harboring the *ftsQ1* allele (EC433) upon exposure to a wide pH range under permissive (left) and non-permissive (right) conditions. C) Table summarizing suppression and enhancement data for strains harboring temperature sensitive variants in late division proteins (EC433, *ftsQ1*; WM2101, *ftsK44*; WM4649, *ftsI23*) across a range of pH conditions. ++, +, and −denote complete, partial, or no suppression at the indicated pH. **, *, and −denote complete, partial, or no enhancement at the indicated pH. (TIF)

**S6 Fig. Production of GFP-tagged division proteins does not eliminate pH-dependent changes in cell length.** A-E) Cell length distributions of cells overexpressing tagged division proteins, including FtsZ-GFP (A, BH330), GFP-FtsA (B, EAM410), GFP-FtsL (C, PAL3700), GFP-FtsI (D, EAM412), and FtsN (E, EAM621) during steady state growth in LB medium at pH 5.5, 7.0, and 8.0. Cells were collected for imaging at $OD_{600} \sim 0.1$–$0.2$. (TIF)

**S7 Fig. Midcell intensity of GFP-tagged late division proteins across pH conditions.** A-C) Mid-cell intensity quantifications (right) and demographs (left) for cells producing GFP-FtsN (A, EAM621), GFP-FtsI (B, EAM412), or GFP-FtsL (C, PAL3700). (TIF)

**S8 Fig. Septal ring frequency of FtsZ-GFP (BH330), GFP-FtsA (EAM410), GFP-FtsL (PAL3700), and GFP-FtsI (EAM412) across a wider pH range.** (TIF)

**S9 Fig. Production of FtsN does not vary across pH conditions.** A) Uncropped membrane shown in Fig 3 probed with anti-MBP-FtsN sera (top) and anti-FtsZ sera (middle) or strained with Ponceau reagent for total protein levels (bottom). Arrow indicates degradation or processed FtsN band. B) Quantification of relative FtsN and FtsZ levels as a function of pH. Bars depict mean relative levels of each protein ± SD relative to pH 7.0 from three independent cultures and normalized for total protein load as determined by Ponceau stain. C) FtsN degradation product as a percentage of total FtsN across pH conditions. (TIF)

**S10 Fig. Production of GFP-FtsN does not vary across pH conditions or strain background.** A) Western blot for GFP-FtsN levels (EAM621) from cells grown to steady state in LB medium at pH 5.5, 7.0, and 8.0. Three replicates for each pH condition are shown. B) Western blot depicting GFP-FtsN levels from MG1655 (EAM621), *ftsA** (EAM747), and *ftsL** (EAM749) grown to steady state in LB medium (pH 7.0). Three biological replicates are shown for each strain. (TIF)

**S11 Fig. *ftsN* overexpression suppresses the heat sensitivity of late division protein variants and bypasses the essential function of FtsK.** A) Representative plating efficiency for cells producing heat sensitive variants of division proteins (PAL2452, *ftsZ84*; WM4107, *ftsA27*; WM2101, *ftsK44*; EC433, *ftsQ1*; WM4649, *ftsI23*) under non-permissive growth conditions in the presence (right) or absence (left) of *ftsN* overexpression (pCH201; 1 mM IPTG). B) MG1655 can grow in the absence of FtsK (EAM1311) upon *ftsN* overexpression (1 mM IPTG). (TIF)

**S12 Fig. Impact of late division protein overproduction on cell length and growth.** A-B) Cell length of MG1655 producing excess FtsN (*pBAD33-ftsN*; A) or FtsI (*pBAD18-ftsI*; B) during steady state growth in LB medium. Cells were collected for imaging at $OD_{600} \sim 0.1$–0.2. Bars represent mean cell length ± SEM from three independent biological replicates (n > 200 cells per replicate). C) Representative growth curves for WT (MG1655) cells +/- *ftsN* overexpression plasmids during growth in LB medium or AB minimal medium + 0.2% glycerol. Cells were grown to steady state in LB medium (uninduced) then inoculated into a 96-well plate in the indicated medium with and without inducer.
(TIF)

**S13 Fig. Production of GFP-FtsN variants.** A-B) Representative Western blots for GFP-FtsN truncations (A) or point mutants (B) expressed in MG1655 during steady state growth in LB medium (+1 mM IPTG) and probed with anti-GFP.
(TIF)

**S14 Fig. *ftsN* depletion across pH conditions.** A) Representative plating efficiency for *ftsN* depletion in WT (HSC074/pBAD33-ftsN), *ftsA\** (EAM719/pBAD33-ftsN), and *ftsL\** (EAM723/pBAD33-ftsN) cells at pH 5.5 (bottom) or neutral pH (right) across induction conditions. Image is representative of three biological replicates. B) Representative plating efficiency for temperature-dependent *ftsN* depletion in WT (MG1655/ *Psyn135::ftsN*) at pH 5.5 (bottom) or neutral pH (right). Image is representative of three biological replicates.
(TIF)

**S1 Table. Bacterial strains and plasmids used in this study.**
(PDF)

**S2 Table. Impact of pH on cell dimensions of MG1655 in LB medium.**
(PDF)

**S3 Table. GFP septal ring frequencies across pH conditions in LB medium.**
(PDF)

**S4 Table. Impact of *ftsN* overexpression on cell size in LB medium.**
(PDF)

## Acknowledgments

We thank David Weiss, Thomas Bernhardt, Piet de Boer, Bill Margolin, and Joe Lutkenhaus for kind gifts of strains, plasmids, and antibodies necessary to carry out this work. We thank David Weiss and members of the Levin lab for helpful discussions, as well as Stephen Vadia for critical reading of the manuscript.

## Author Contributions

**Conceptualization:** Elizabeth A. Mueller, Corey S. Westfall, Petra Anne Levin.

**Formal analysis:** Elizabeth A. Mueller.

**Funding acquisition:** Elizabeth A. Mueller, Petra Anne Levin.

**Investigation:** Elizabeth A. Mueller.

**Methodology:** Elizabeth A. Mueller, Corey S. Westfall, Petra Anne Levin.

**Resources:** Elizabeth A. Mueller, Petra Anne Levin.

**Supervision:** Petra Anne Levin.

**Visualization:** Elizabeth A. Mueller.

**Writing – original draft:** Elizabeth A. Mueller, Petra Anne Levin.

**Writing – review & editing:** Elizabeth A. Mueller, Corey S. Westfall, Petra Anne Levin.

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
