## [Decision Letter · Decision Letter 0]

25 Oct 2019

Dear Petra,

Thank you very much for submitting your Research Article entitled 'Environmental pH impacts division assembly and cell size in Escherichia coli' to PLOS Genetics. Your manuscript was fully evaluated at the editorial level and by independent peer reviewers. The reviewers appreciate the attention to an important problem, but agree that some significant modifications are required to improve this manuscript. In particular, reviewer 1 raised substantial concerns regarding the data implicating FtsN in pH-responsiveness and provides suggestions on how these issues can be addressed experimentally and by modifying the text. Based on the reviews, we will not be able to accept this version of the manuscript, but we would be willing to review again a much-revised version. We cannot, of course, promise publication at that time.

If you decide to revise the manuscript for further consideration at PLOS Genetics, please aim to resubmit within the next 60 days, unless it will take extra time to address the concerns of the reviewers, in which case we would appreciate an expected resubmission date by email to plosgenetics@plos.org.

[LINK]

We are sorry that we cannot be more positive about your manuscript at this stage. Please do not hesitate to contact us if you have any concerns or questions.

Yours sincerely,

Kristina Jonas, Ph.D.

Guest Editor

PLOS Genetics

Gregory P. Copenhaver

Editor-in-Chief

PLOS Genetics

Reviewer's Responses to Questions

**Comments to the Authors:**

Reviewer #1: In this manuscript, Mueller and colleagues establish and explore mechanistically the relationship between extracellular pH and cell size in Escherichia coli. They find that cell length is correlated with extracellular pH (lower pH = shorter cells, higher pH = longer cells) implying that cell division is responsive to extracellular pH. They use genetics and imaging to implicate the late-arriving division protein FtsN as a potential mediator of this response. FtsN localizes more frequently and robustly to the division site at low pH than at high pH. In addition, overexpression of ftsN or ftsN truncation variants that include the cytoplasmic domain and CCD is sufficient to induce cell shortening at neutral pH. Hyperactivating mutants in ftsA or ftsL previously shown to bypass essentiality of ftsN render cell division insensitive to pH, though FtsN is more robustly recruited to the division site in those mutants. Collectively, their data lead the authors to propose that in E. coli, acidic pH leads to hyper-recruitment of FtsN, which then leads to activation of the division machinery. They further propose positive feedback from division activation to recruitment of FtsN to robustly trigger cell division.

Overall, the manuscript is clearly presented and, for the most part, logical. The conclusion that activation of cell division is responsive to extracellular pH is well-supported and likely to be of broad interest. However, the conclusion that FtsN is the critical mediator of this regulation in E coli is not sufficiently supported by the data presented. My specific comments are as follows:

Major:

1. The implication of FtsN in mediating pH-responsiveness of the division machinery came initially from localization analysis of 5 division proteins (FtsZ, FtsA, FtsL, FtsI, and FtsN) at varying pH. Ideally localization of other late division proteins should be tested, as well (i.e. FtsW and FtsQ), to gain confidence in the specificity of FtsN enrichment at low pH. Although FtsN is canonically thought to be the last recruit to the divisome in E coli, following FtsI, alternative routes to FtsN recruitment have been suggested under some conditions. Moreover, from supplementary figure S7, GFP-FtsI appears moderately enriched at midcell at lower pH, as well. In addition, it would be useful to have the same full set of data/data analysis presented for each protein imaged either in a main figure or supplement (i.e. for FtsZ, there are images shown in Figure 3D but no demograph or fluorescence quantification as in Figure S7 and, conversely, for FtsI no cell images are shown, but demograph and fluorescence quantification is included). Finally, in the images shown in Figure 3D and E, the total fluorescence appears to decrease as pH increases for both FtsZ-GFP and GFP-FtsN. Is that the case?

2. The second piece of evidence implicating FtsN in pH-responsiveness of division is the observation that overexpression of ftsN or ftsN truncations encoding the cytoplasmic region and CCD cause cell shortening at neutral pH. The authors note that two other previous reports of ftsN overexpression in E. coli did not observe cell shortening and actually reported toxicity and cell lengthening. The authors do not discuss why their results might differ from previously published work. Given that this is the only functional evidence presented in the current manuscript directly linking FtsN enrichment to divisome activation, this requires further elaboration. Did the authors try to acquire the constructs used in prior work and test them in their hands? Did they test overexpression of untagged ftsN truncation/mutant variants?

3. The authors demonstrate that E. coli bearing hyperactivating mutants of ftsA and ftsL that bypass essentiality of ftsN do not change their length in response to pH. In those strains, GFP-FtsN is hyper-enriched at midcell. This led the authors to suggest positive feedback from division activation to FtsN recruitment, which makes sense. However, this brings the chicken-or-egg problem to the pH responsiveness, as the authors note on page 12. Is GFP-FtsN enriched at midcell at low pH, leading to division activation as the authors propose? Or is the divisome hyperactivated at low pH through an FtsN-independent mechanism, causing enrichment of GFP-FtsN at midcell through positive feedback? The authors try to test this by determining if lower levels of FtsN might be sufficient for growth at low pH, as they are in ftsA* and ftsL* mutants, and find that they are not (at least at the arabinose concentration tested - might testing intermediate concentrations be informative?). This suggests that low pH does not hyperactivate division in the same way the ftsA or ftsL mutants that bypass ftsN do, but given the lack of mechanistic detail of the activation pathway for division, a negative result here is insufficient to rule out the possibility that division is hyper activated at low pH through a mechanism independent of FtsN enrichment. Given the broad conservation of pH-responsiveness of cell division, but lack of broad conservation of FtsN, an FtsN-independent mechanism of pH responsiveness seems likely (perhaps working in concert with low pH-mediated FtsN enrichment in E. coli).

4. A number of conclusions are overstated:

a. lines 34-36: “…environmental pH impacts the length at which cells divide by altering the ability of the terminal cell division protein FtsN to localize to the cytokinetic machinery and activate division.” The localization is fairly well-supported, but there is no evidence presented that supports the conclusion that pH impacts the ability of FtsN to activate division. Do the authors mean “to localize to the cytokinetic machinery where it activates division”?

b. lines 282 and 927: “Acidic pH activates division through FtsN”. This section and figure title are overstated. The authors show that ftsA and ftsL hypermorphs are not responsive to pH, that FtsN is enriched at midcell in those backgrounds, and that FtsN is not dispensable at low pH. These data are not sufficient to draw the conclusion that low pH activates division through FtsN.

c. line 310: “…our data indicate FtsN is necessary and sufficient for low pH-mediated division activation…”. The overexpression suggests that FtsN is sufficient to induce division hyper-activation, but as that experiment was performed at neutral pH it does not show that FtsN is sufficient to activate division at low pH. Necessity would derive from non-responsiveness of cells lacking FtsN to pH, but as ftsN is apparently essential for growth at all pHs, this cannot be directly assessed.

Other specific points:

5. line 153: “In the final phase of division in proteobacteria, FtsN accumulates at mid-cell and is believed to “trigger” septal PG synthesis…” This has only been shown in E. coli. Though FtsN is conserved in proteobacteria, in Caulobacter it is recruited well before FtsW or FtsI. I know of no evidence in organisms outside of E. coli that implicates FtsN as a trigger for division.

6. line 217: “Because populations of E. coli cell…” should be “E. coli cells”

7. line 248-9. Which strain in Table S4 is this section referring to? Was this supposed to be Table S5?

8. Related to point 7 - it would be really helpful to have brief strain descriptions in each of the tables (e.g. Table S4, S5) to allow the reader to quickly determine what is being expressed in each case. In addition, having relevant strain numbers listed in the figure legends to allow the reader to determine what strain is being used in each experiment would be helpful.

9. line 313: “… influence in its septal…” should be “influence on”

10. lines 337-338: It seems relevant to cite reference 20 here, as they demonstrated an effect of division hypermorphs on cell size homeostasis.

11. Figure 5 legend - there is no legend for 5C, and the legend for 5D is mislabeled as C. At what pH was GFP-FtsN localization assessed in 5C?

12. Figure 6. Label on the Y-axis of the graph in A should be “septal” not “septa”

13. Supplemental tables S1 and S2. Arranging strains and plasmids alphabetically by name would help with ease of reference.

14. Tables S3 and S5. Please define “MDT” in the legend.

Reviewer #2: This well written manuscript of Mueller et al. describes the effect of medium pH on cell division and pin-point the activity to FtsN in an intelligent stepwise genetic dissection approach. The importance resides in the fact that they show that cell size is not simply determined by growth rate, and that several cell division mutants can be bypassed or enhanced by changing the pH. This will facilitate the research in cell division since it opens new avenues to create conditional lethal mutations, a key tool in the investigation of bacterial cell division.

The results are clearly presented and build up logically, except for the last two figures, which might be more logical to reverse. Also, I do not necessarily agree with some of the sub-conclusion they pose (see details below). One additional result that I completely cannot understand is how the SPOR domain by itself (GFP-FtsN(243-319)) can localize at the cell division site (Figure 6A) since this domain would than need an N-terminal signal peptide to get transported into the periplasm. Maybe I missed something, but the authors have to make clear how this is possible.

Detailed comments:

l.59: , size, is

l.71: added

l.83-86: it would be reasonable to add the effect of pyruvate kinase in B. subtilis (Monahan, 2014, MBio).

l.95: as divisome

l.122: size iso area?

Indicate in S3 what MDT is

l.135: maybe mention why W3110 was chosen?

l.225: This conclusion is not correct since the localization of FtsA increases slightly but significantly in Fig. 3B

l.232: in Figure S7 there is also a slight enhancement in FtsI?

l.244: you cannot enhance an interaction by overexpression (you can increase the chance of interaction).

Since Figure 4 discusses the truncation study, the question immediately arises how these truncations behave at different pHs. Therefore it might make more sense to follow with the results that are now presented in Figure 6A. This might make the reasoning also easier (see comments below).

l.286: This reasoning is confusing. Firstly, the explanation stated here is not in line with the two models listed on page 10 line 234-236. Secondly, the suggested activation of FtsA by FtsN has not been mentioned before. Only that FtsN interacts with FtsA. The activation, which is not the same as binding, of FtsQLB, has also not been mentioned earlier on. In any case, because it is not known how those hypermorphic FtsA and FtsL mutants bypasses the need for FtsN it is very hard to draw clear conclusions from negative results, i.e. no effect of pH in these mutants.

l.298: The model that pH regulates FtsA does not make sense since it is dismissed on forehand as it is a cytoplasmic protein, as is also stated in line 323.

l.316: How does GFP-FtsN241-319 gets out of the cell without a signal sequence?

In the Discussion it would be good to mention that FtsN is not present in Staphylococci, so that other late proteins must be affected in this organism.

l.410: references?

l.411-413: Where is the evidence for the first conclusion? Not in Figure 5.

Reviewer #3: This is a well-written and clearly presented study on the effects of environmental pH on cell size in E. coli, showing that growth in acidic pH results in smaller cells compared to neutral and basic pH growth conditions, irrespective of growth rate. Such behavior strongly points towards enhanced cell division at lower pH. The authors present compelling evidence for the enhanced accumulation of FtsN at division sites at low pH as the mechanism underlying the observed reduced cell length. The enhanced localization is dependent on the presence of FtsN’s cytoplasmic FtsA-binding and periplasmic CCD domains, and can partially suppress or totally bypass the essentiality of late division proteins, including FtsI, FtsQ, and FtsK. Although the actual mechanism that triggers enhanced FtsN recruitment by low pH conditions remains unclear, this study contributes more to our understanding of the mechanistic regulation and assembly of the divisome and how extracellular factors can perturb the balance between division and cell size control. However, the manuscript requires some modifications, as indicated below.

Major comments:

1. Given the authors’ previous work on the differential fitness contributions of the Class A PBPs in acidic and basic media, and the recent publication (PMID: 30504892) linking FtsN to the PBPs (which should certainly be cited), the authors need to include some discussion on how their interpretation of their data fits into the previous studies.

2. Fig. 4D is confusing. The histogram actually shows the % of GFP-FtsN at the septum, not FtsN. The % of FtsN at the septal ring is not 0 in WT cells, as it must already be at the septum in order for them to divide. A better measure would be IFM using Anti-FtsN, or to change the Y axis to “GFP-FtsN” and to state more clearly that this is a measure of how much GFP-FtsN localizing to the septum, not total FtsN. The same is true for Fig. 5C.

3. The legend for figure 5 needs to be corrected; the description of panel C is missing and the description for panel D is mislabeled.

4. Can the authors explain why they see stimulation of division by excess FtsN (or GFP-FtsN) whereas previously this was not seen (or excess FtsN inhibited division)? The WT (or uninduced) cell length of >4 µm seems high. Could that be why?

5. Supplementary Figure 4. The authors attempt to show that the pH-dependent changes in cell length are due solely to FtsN by demonstrating that cells deficient in other periplasmic pH-responsive proteins show the same phenotype. However, in their previous publication (ref. 8), ΔmrcB cells did not grow in acidic media and eventually lysed. It seems that this is an inappropriate conclusion to draw from this experiment based on what the authors reported previously.

6. Figures 6A and 4E; lines 321-329. Based on their truncation mutants, the authors conclude that the FtsA-binding domain and CCD domain of FtsN are both required for the cell-shortening phenotype, as well as localization to the septum. They reason that the FtsA binding domain is less important in this interaction since the truncation mutant 1-81 does not exhibit pH-dependent localization. However, this conclusion may be erroneous, as GFP-FtsN(1-55) localizes quite efficiently to the septum as reported in ref. 60. Is the GFP used here different? Or perhaps the region between 55 and 81 of FtsN inhibits its septal localization?

7. The title should be reworded—“division assembly” is an ambiguous term. I suggest replacing “division assembly” with something more specific, such as “cell division”, “division septum formation”, or “cytokinesis”. I am also not a fan of using “impacts” as a verb, but realize that it is now in common usage.

Additional comments:

1. Figure 2: In panel A, the temperature sensitive allele of FtsA is labeled “ftsA12”; however, in the text the primary ts allele of FtsA referenced is FtsA27 (line 180), with FtsA12 being mentioned for the supplementary figure S5 (line 186).

2. Line 192: Fig. 2B is cited here, but it does not show data for ftsA27 or ftsZ84 as stated.

3. Figure 5D. Can the authors comment on why there seems to be a 10-fold increase in growth of WT cells when depleted for FtsN at pH 8.0 compared to pH 5.5 or pH 7.0?

4. Figure 5C; lines 291-292. The authors state that the frequency of GFP-FtsN localization at the septum was similar between FtsA* and FtsL* cells, but it is not clear if they tested or showed this at each pH condition.

5. The “inactive state of divisome proteins in discrete complexes” (lines 400-405) model proposed is similar to that described in Krupka et al. (PMID: 28695917), who proposed that FtsA is maintained in a “locked” mini-ring state prior to divisome activation based on direct visualization of oligomeric structures of FtsA and FtsA* on membranes. That paper also proposed a positive feedback loop among early divisome proteins to activate cytokinesis. The authors should mention this as a possible mechanism, especially given that transition of FtsA from the OFF to ON state is contributing to their model.

6. Lines 237-238: It looks like levels of GFP-FtsN are lower at high pH compared with neutral pH in both Fig. 3E (S8) and Fig. S6. There is also a more prominent degradation/processed band at higher pH. This may or may not be a factor in the sensitivity to high pH, and should be mentioned.

7. Line 249-250: Do the authors have any thoughts as to why cell width increases upon induction of GFP-FtsN?

8. Line 286: The original paper that reported shorter cell lengths from hypermorphic alleles of cell division genes should be cited as well (PMID: 17322202).

9. Line 296: Does the enhanced recruitment of FtsN to the septum cause hyperactivation of FtsA/FtsQLB, or mimic their hyperactivation through some other mechanism? This would seem to be an important distinction.

10. Line 302: The first report that certain alleles of FtsA including FtsA* can survive without functional FtsN should be cited here (PMID: 17542921).

Typos:

Line 217: should be “cells”

Line 313: delete “in”

Figure 2A: “fsK44” should be “ftsK44”.

**Have all data underlying the figures and results presented in the manuscript been provided?**

Reviewer #1: Yes

Reviewer #2: Yes

Reviewer #3: Yes

PLOS authors have the option to publish the peer review history of their article (what does this mean?). If published, this will include your full peer review and any attached files.

Reviewer #1: No

Reviewer #2: No

Reviewer #3: No

---

## [Decision Letter · Decision Letter 1]

17 Feb 2020

Dear Petra,

Thank you very much for submitting a revised version of your Research Article entitled 'pH-dependent activation of cytokinesis modulates Escherichia coli cell size' to PLOS Genetics. We have returned your manuscript to reviewer 1, who was satisfied with the changes that you have made, but has suggested a few minor text changes - please address these and we'll be ready to render a decision without further external review.

We therefore ask you to modify the manuscript according to the review recommendations before we can consider your manuscript for acceptance. Your revisions should address the specific points made by the reviewer.

1) Provide a list of your responses to the review comments and a description of the changes you have made in the manuscript.

[LINK]

Yours sincerely,

Kristina Jonas, Ph.D.

Guest Editor

PLOS Genetics

Gregory P. Copenhaver

Editor-in-Chief

PLOS Genetics

Reviewer's Responses to Questions

**Comments to the Authors:**

Reviewer #1: The authors have responded appropriately to my prior concerns. The study is nicely done, and the manuscript is logically and clearly presented. I have only a couple of minor changes to suggest.

1. Lines 136 and 142: "evolutionary distant" should be "evolutionarily distant"

2. Line 223: "differences in in midcell localization" has an extra "in"

3. Line 233: should the figure reference be to Figure 3D, not 3E?

4. Also line 233: "SI Appendix, S8" should be "SI Appendix, Fig. S8" I think

**Have all data underlying the figures and results presented in the manuscript been provided?**

Reviewer #1: Yes

PLOS authors have the option to publish the peer review history of their article (what does this mean?). If published, this will include your full peer review and any attached files.

Reviewer #1: No

---

## [Editor Report · Decision Letter 2]

19 Feb 2020

Dear Petra,

We are pleased to inform you that your manuscript entitled "pH-dependent activation of cytokinesis modulates Escherichia coli cell size" has been editorially accepted for publication in PLOS Genetics. Congratulations!

Yours sincerely,

Kristina Jonas, Ph.D.

Guest Editor

PLOS Genetics

Gregory P. Copenhaver

Editor-in-Chief

PLOS Genetics

Comments from the reviewers (if applicable):

**Data Deposition**

http://datadryad.org/submit?journalID=pgenetics&manu=PGENETICS-D-19-01602R2

**Press Queries**

---

## [Editor Report · Acceptance letter]

13 Mar 2020

PGENETICS-D-19-01602R2 

pH-dependent activation of cytokinesis modulates Escherichia coli cell size 

Dear Dr Levin, 

We are pleased to inform you that your manuscript entitled "pH-dependent activation of cytokinesis modulates Escherichia coli cell size" has been formally accepted for publication in PLOS Genetics! Your manuscript is now with our production department and you will be notified of the publication date in due course.

With kind regards,

Kaitlin Butler

PLOS Genetics

On behalf of:
